# Maternal Depression: Relationship to Food Insecurity and Preschooler Fruit/Vegetable Consumption

**DOI:** 10.3390/ijerph17010123

**Published:** 2019-12-23

**Authors:** Wendy L. Ward, Taren M. Swindle, Angela L. Kyzer, Nicola Edge, Jasmin Sumrall, Leanne Whiteside-Mansell

**Affiliations:** 1Department of Pediatrics, College of Medicine, University of Arkansas for Medical Sciences, Little Rock, AR 72202, USA; 2Department of Family and Preventive Medicine, College of Medicine, University of Arkansas for Medical Sciences, Little Rock, AR 72202, USA; TSwindle@uams.edu (T.M.S.); ALKyzer@uams.edu (A.L.K.); NAEdge@uams.edu (N.E.); WhitesideMansellLeanne@uams.edu (L.W.-M.); 3Harding University, Searcy, AR 72202, USA; jsumrall@harding.edu

**Keywords:** maternal depression, food insecurity, fruit/vegetable consumption, pediatric obesity, early childhood, child development

## Abstract

*Background*: Maternal Depression (MD) has been implicated in the etiology of obesity. The present study investigated MD and both child fruit/vegetable consumption (FVC) and household food insecurity (FI) in an early childhood population. *Methods*: This cross-sectional study was conducted in Arkansas, United States, in 26 Head Start centers. Teachers obtained the Family Map (FM), an interview assessment tool used by Head Start staff to identify potential risk factors that affect child learning and development. The FM contains a two-item screener of parent depression—the Patient Health Questionaire-2, two questions about family FI, and two questions about FVC. The FM was completed in 693 households. Chi square analyses and logistic regressions utilizing adjusted and unadjusted odds ratios were utilized to compare differences in risk for children of mothers with no symptoms compared to mothers with low- or high-level depressive symptoms. *Results*: Children whose mothers had high MD were 2.90 (CI: 1.21–7.00) and 7.81 (CI: 3.71–16.45) times more likely to be at risk for low FVC and FI, respectively, compared to children of mothers with no MD. Similar findings but at lower magnitude were found for mothers with low symptoms of MD in comparison with mothers with no MD—both for Low FVC (1.57 times more likely; CI: 1.01–2.45) and FI (2.14 times more likely; CI: 1.28–3.58). The results presented are Odds Ratios from the multivariable adjusted models. *Conclusions*: Implications for the etiology of obesity, prevention/intervention efforts, and future research are offered, including recommended addition of maternal depression and household FI screening in early childhood programs.

## 1. Introduction

Prior research has documented associations between maternal depression and underweight status and poor growth trajectories in infants and toddlers [1,2,3]. Recently, maternal depression has also been linked with childhood obesity [4,5]. This was a novel consideration that received national attention. A recent four-wave longitudinal study tracked children from early childhood through preadolescence and found maternal depression when the child is in kindergarten predicted significant increases in body mass index (BMI) in fifth grade boys and girls [5]. Another recent study found that children of depressed mothers were about twice as likely to become overweight as children of non-depressed mothers [6]. Given these results, an examination of the mechanism(s) underlying the relationship between maternal depression and child obesity is warranted.

A wide range of research supports the finding that parenting effectively is difficult for caregivers with depressive symptoms. For example, in a review of maternal depression and parenting behavior, researchers found that maternal depression has strong associations with negative maternal behavior and moderate associations with disengagement from the child [7]. Depressed mothers exhibit higher levels of hostility and negative interactions with their children [8], impatient use of directives in guiding child behavior [9], and less positive interactions with their children [10]. Numerous negative outcomes have been found for children of depressed mothers as well, including internalizing and externalizing disorders [10,11]. Maternal depression is linked specifically to the mother’s inability to parent effectively or consistently [7,11,12], including problematic feeding practices [13,14,15]. These practices include low parental control of availability and access to unhealthy foods [15], parent modeling of negative food-related behaviors [15], and low vegetable intake [16].

Negative parenting practices are particularly pronounced in socioeconomically disadvantaged households [7,10] who have food insecurity [17]. Food insecurity (FI) is defined as the absence of “consistent, dependable access to enough food for active, healthy living. [18]” Prevalence rates suggest that 10%–15% of United States households are food-insecure [18,19]. Many studies have found links between FI and poor child health outcomes, including obesity [4,17,20], malnutrition [21], physical and mental health outcomes [22], and cognitive and motor delays [22,23]. FI is associated with a lower consumption of fruits and vegetables [24], suggesting poor food quality [25]. Most importantly to this discussion, FI is associated with greater mental health problems in mothers [4,26], including maternal depression [25].

Early childhood may be an important time to investigate the relationship between maternal depression and obesity. Developmentally, the role of the mother is central to children’s physical and psychological status in infancy and early childhood. Theorists suggest that maternal depression may be particularly detrimental to young children given the central and important role of the caregiver [10]. Furthermore, prevalence of overweight and obesity is high even in early childhood [27,28]. There is increased interest in the link between early indicators of social risk (such as FI and maternal depression) and later health risks such as obesity [29]. A proposed conceptual model representing these links is presented in Figure 1.

This study was instigated by the emerging conversation in pediatric obesity circles about the importance of maternal depression. The extant literature suggests that maternal depression has both a direct and indirect link with child obesity through the factors of food insecurity and dietary intake. This figure represents the foundation of the present cross-sectional study in which we test the associations between maternal depression, FI, and dietary intake (i.e., fruit and vegetable consumption). This study targets early childhood in families with few financial resources. Specifically, we expect that maternal depression symptoms will be associated with lower fruit and vegetable consumption (FVC) and greater risk of FI.

## 2. Methods

### 2.1. Sample and Procedures

This study was conducted in 2006 in the state of Arkansas in the United States. While FI has been found to be significantly related to both undernutrition and obesity in the literature, the incidence of the latter is of greater magnitude in the state of Arkansas, which ranks as one of the top three in obesity rates in the United States. In fact, obesity rates have been documented to be higher among individuals in our state with food insecurity (42%) than among food secure individuals (33%) [30].

Most centers were located in a metropolitan area (50,000+ population; 20 Head Start centers, *n* = 53 classrooms). Fewer were in areas serving families living in a micropolitan area (<50,000 population; 6 centers, *n* = 17 classrooms). Head Start is a federally funded program serving families and their children (ages 3 to 5) whose incomes are at 100% of poverty or below. Head Start is a two-generational model that aims to strengthen families as the primary nurturers of children [31]. Head Start exists to support the healthy development of children through the provision of supportive interventions, educational services, and referrals to community services. The Family Map Inventories [32,33] are standard assessment tools that can be implemented in home visits and assists Head Start personnel in identifying risk.

In the present study, all parents in each class were interviewed as part of the Head Start program by educators using the Early Childhood Family Map Inventory. Each mother was interviewed once for each of her children in Head Start but only one child (the oldest) was included in the dataset (*n* = 67 siblings removed. A total of 1035 male and female participants completed the assessment. For the purposes of this study, however, analyses only included caregivers who indicated their gender as “female,” as we were specifically examining maternal depression. Caregivers (*n* = 771) were interviewed in their home by a Head Start educator. A small number of respondents (*n* = 78, ~10%) were not included in the analyses due to missing data on variables of interest. Thus, our final sample consists of 693 caregivers, including both biological mothers (94%) and mother figures, such as grandmothers and foster mothers. The Institutional Review Board at the University of Arkansas for Medical Sciences approved this study, and the authors conducted this work in accord with prevailing ethical principles.

### 2.2. The Family Map Inventory

The Family Map Inventory is conducted in a semi-structured interview format. Questions in the interview assess factors in 12 domains of the family and home environment known to influence the well-being of young children. The Family Map Inventory is designed to be culturally relevant for this population [32]. Questions identify areas of concern and strengths for families/children to inform interventions to promote conditions associated with healthy development and to reduce environmental risks. The Family Map often takes places during home visits with Head Start families (length varies but is most often 60–90 min). The use of the Family Map required 6 h of standardized training.

Reliability and validity of the Family Map constructs have been provided in detail elsewhere [32,34]. Risk estimates from the Family Map are consistent with national estimates (e.g., Family and Child Experiences Survey) and estimates found in other studies, which supports the validity of the Family Map [32,33]. Family Map constructs have also been documented to correlate as expected with conditions in the home as captured by direct observation and by other standard instruments in the field [25,34,35], offering additional validity support. Modules relevant to this study are described below, along with available reliability data.

#### 2.2.1. Maternal Depression

Depression in the Family Map is assessed using the Patient Health Questionnaire-2 (PHQ-2) [32,35,36]. PHQ-2 items are recommended by the U.S. Preventive Services Task Force as a valid and efficient screening option [37]. Summary scores on the two-item PHQ-2 range from 0–6. With a cut-off score of 3, a recent systematic review shows that the PHQ-2 has a sensitivity for major depression of 83% and a specificity of 92% [38]. The PHQ-2 has been used as a screen for maternal depression for the purposes of intervening with child outcomes [39]. Questions are: In the past 2 weeks, how often have you been (1) Bothered by feeling down, depressed, or hopeless and (2) Bothered by having little interest or pleasure in doing things.

Responses and scoring for each item include: not at all (0), several days (1), more than ½ the days (2), and nearly every day (3). For purposes of this study, we summed these responses, creating three maternal depression groups: (1) Caregivers with no depressive symptoms (PHQ-2 = 0), (2) Low-level depressive symptoms as a score of 1 or 2, and (3) high-level symptoms scoring 3 or more. We use the term “maternal depression” for simplicity, although we are focusing on symptoms rather than a diagnosis.

#### 2.2.2. Fruit/Vegetable Consumption

FVC was assessed using the Family Map with a dichotomous variable. FVC was created using a cut above or below recommended daily intake [32]. Parents responded to two questions which were used to define risk “How often does your child eat a food from each of the following food groups: (a) Dark green or orange/yellow vegetables such as greens, carrots, broccoli, squash, sweet potatoes—but not French fries, and (b) Fruits like apples, oranges, bananas, grapes, peaches, applesauce—but not juice.” Parents indicated response on a 1 to 5 scale, where 1 = None, 2 = Once a week, 3 = 2–6 times a week, 4 = Once a day, and 5 = More than one per day. Children eating one or more fruits AND one or more vegetables per day were designated as above the minimum consumption risk [32]. This measure has been used successfully to investigate FVC in early childhood [32,35], and test-retest reliability for this module is 68% [32].

#### 2.2.3. Food Insecurity

FI is assessed in the Family Map with two items from the United States Department of Agriculture’s U.S. Household Food Security Survey Module: Six-Item Short Form. [25,32]. The FI assessment has been validated with findings supporting convergent validity, sensitivity, and specificity [25]. This module has an established test-retest reliability of 68% [32]. Participants were asked how often in the past year: (1) “The food that you bought just didn’t last and you didn’t have money to get more,” and (2) “You or others in your household cut the size of your meals or skipped meals because there wasn’t enough money for food.” Participants indicated if the two FI items were (1) never, (2) sometimes, or (3) often true for their family. Families responding “sometimes” or “often” to either item were designated as food insecure. Internal consistency in this sample was 0.74 [25].

### 2.3. Analysis Plan

All analyses were conducted using SPSS (SPSS Inc., Chicago, IL, USA). Preliminarily analyses used chi-square and ANOVA to compare maternal depression groups. Chi-square tests investigated dependence between maternal depressive symptoms and risk in child FVC or household FI at the bivariate level. This analysis compared mothers with no depressive symptoms, low-level depressive symptoms (PHQ-2 = 1 or 2), and high-level depressive symptoms (PHQ-2 of ≥3) to identify differences in FVC and FI for children. Logistic regression models with controls for family demographics were used to investigate the differences in risk for children of mothers with no symptoms compared to mothers with low- or high-level depressive symptoms. Control variables included the presence of another adult in the home, child gender, maternal race/ethnicity, level of maternal education, employment status, and number of children in the home. All control variables were also captured through the Family Map interview.

## 3. Results

### 3.1. Maternal Depression

Table 1 provides the distribution of responses to the PHQ-2 items. Consistently with the general population, the majority of mothers in the sample were not depressed (72.0%); approximately one quarter of mothers had low-level depressive symptoms (22.2%); and an even smaller group had high-level symptoms (5.8%).

### 3.2. Demographics

Sample demographics are shown in Table 2 by maternal depression level. The majority of caregivers were black (55.7%) followed by white (22.2%), Hispanic (14.3%) and other (7.8%). Most caregivers were employed (67.5%) and had a partner (57.2%). Half had a high school degree or less, and half had some post-secondary education. Many caregivers were the only adult in the home (38.5%), with an average of 2.23 (SD = 1.14) children in the home. About half (46.6%) of children were male.

The maternal-depression groups are similar in terms of racial composition, presence of another adult living in the home, marital status, and educational level (see Table 2). There were significant group differences, however, in two areas: (1) employment (χ^2^ = (2, 679) = 11.54, *p* = 0.003), with mothers in the ‘no-symptom’ group being the most likely to be employed; and (2) number of children living in the home (*F* (2, 681) = 7.40, *p* = 0.001), with ‘no symptom’ mothers having the fewest children.

### 3.3. Child Fruit and Vegetable Consumption and Food Insecurity

Most children (67.2%) had inadequate FVC in our sample, and 16.1% were food insecure. Unadjusted and adjusted odds ratios showing the relationship between maternal depression symptoms and FVC and food insecurity are shown in Table 3. Adjusted for demographics, the child of a mother with low-level symptoms of depression was 1.5 times more likely to experience deficits in FVC, and the child of a mother with high-level symptoms was almost three times more likely to experience deficits in FVC compared with families where mothers exhibited no depressive symptoms (see Table 3). Households in which mothers had low-level depressive symptoms were more than twice as likely to have problems with FI compared to households where mothers had no depression symptoms. When compared to children of mothers with no symptoms, children of highly depressed mothers were almost eight times more likely to be at risk for household (*OR* = 7.81, *CI* [3.71, 16.45], *p* = 0.004).

## 4. Discussion

In this study, children of depressed mothers were more at risk for poor nutrition intake both from lack of any food and from lack of consumption of sufficient fruits and vegetables. While prior research has found a relationship between maternal depression and household FI [4,25,26], findings from this study suggest that maternal depression is also associated with the quality of children’s diets. The link between maternal depressive symptoms and child fruit/vegetable consumption was pronounced in this study, which suggests an important target for intervention.

Although the relationship between maternal depressive symptoms and FVC is striking in this study, caution is needed in interpreting causality. The present study was cross-sectional. It is possible that the variables are interrelated and bidirectional in their effects and, in fact, could represent a dynamic ongoing process. For example, it seems likely that maternal depression’s impact on FVC is an ongoing process [10,12] and might include the foods presented at mealtimes, encouragement/engagement with the child during mealtimes, parental modeling to eat the healthy foods, energy level to shop for groceries or cook [40], or poor parenting behaviors leading to negative child behaviors and resistance [41]. It is also possible that feeling ineffective in getting your child to eat healthy is related to feeling more depressed.

Similarly, maternal depression and household FI may have a bidirectional relationship and may be intertwined as a dynamic process over time. In fact, researchers found that FI predicts increases in depression in women better than income or education [42]. Conversely, women experiencing depression are more likely to be food insecure [43]. Longitudinal research is needed to address the interrelationship between all three variables more clearly—maternal depression, household FI, and child FVC—with additional measures of key constructs, replicating findings in other samples, and tracking changes over time with and without targeted interventions.

There are several implications of these findings from a clinical program perspective. Prevention/intervention programs like Head Start should consider maternal depression and household FI as worthy of assessment and targets for programming. As stated previously, both maternal depression and household FI are related to a wide variety of child outcomes, including health-related behaviors such as fruit/vegetable consumption, but also poor physical health, poor mental health, and delayed development. This study utilizes a maternal depression screener that has a clinical cutoff score to refer for further evaluation and treatment. As maternal depression is identified, diagnosed clinically, and treated, corresponding improvements in child development, parent–child interaction, and health-related behaviors, such as fruit/vegetable consumption, could be tracked.

The present study is not without limitations. From a statistical standpoint, the odds ratios for high depressive symptoms are imprecise given the width of some confidence intervals, attributable to the small number of mothers with high levels of depressive symptomatology at the time of our study. Findings need to be replicated in larger samples, including greater numbers of mothers with low-level and high-level depression. Secondly, the data obtained in this study were provided by self-report, leading to a concern about veracity. Respondents may have provided more socially desirable responses or been reluctant to fully disclose concerns. However, in this study, as in others [32], extremely low FVC and high rates of FI were reported, suggesting some degree of openness and candor [32,35]. Furthermore, replication of results with youth across the age span, in non-poverty samples, in non-Head Start samples, with non-interview formats, and in other geographical locations is also needed. Moreover, the subsample of caregivers with high-level depressive symptoms had a significantly higher prevalence of unemployment than caregivers with low-level depressive symptoms, suggesting a potential confound of the results; namely, that having lower financial resources in general, rather than food insecurity or maternal depression specifically, is related to lower child FVC. Furthermore, in our sample, the caregivers with higher levels of depression had more children in the home than those with low levels or no depressive symptoms. However, the odds ratio analysis was adjusted for both variables (and other potential confounds), and results were still pronounced. It is important to note that the maternal history of psychiatric disorders was not obtained and may increase the risk of both maternal depression and food insecurity.

It is important to note that this study found that mothers with no MD were more likely to be employed and have fewer children. While these results were adjusted for in analyses, they raise the question of whether poverty (more likely if unemployed and/or with more children) is a residual confounding factor. Furthermore, the study participants were recruited from Head Start, a federally funded early childhood education program for children from families with incomes below the federal poverty level who are likely to be under financial stress. In fact, poverty is a common cause (as opposed to confounder) of both maternal depression [44] and inadequate diet [45]. In the U.S., 15% of the population has FI. FI is much higher (34.5%), however, in families below the poverty level [45] and is associated with overweight and binge eating [46]. Further, FVC is strongly connected with diet quality in addition to quantity and FI with undernutrition [2] as well as obesity [2,4]. Thus, future investigations into FVC, FI, and MD should consider socioeconomic status, undereating and underweight, and overeating and overweight variables.

Additionally, while the literature supports cause for concern for children experiencing household food insecurity, little research has been completed on actual child food insecurity. Most data available (including those from administration of the Family Map) utilize tools that look at the household’s food security as a whole rather than focusing on a specific target child and his/her needs or coping strategies. This is worthy of note, because often times, the adults in the household will attempt to protect the children and forgo eating so that the younger children will have nourishment. Researchers found that in households “in which the oldest child was five years old or younger, the prevalence of very low food security among children was less than 0.4% compared with 5.3% for very low food security among adults. [18]” The potential difference between FI as experienced by the child and FI as experienced by households should be held in mind before conclusions are drawn from data on household rather than child food security.

## 5. Conclusions

Maternal depression at any level was associated with an escalated risk for poor child FVC and with increased household FI, with higher levels of both associated with high-level depressive symptoms. Clearly, maternal depression is associated with FVC in early childhood. Furthermore, maternal depression and household FI are strongly related. Future research is needed to better understand the mechanisms underlying these relationships and changes over time. Prevention/intervention programs should assess for maternal depression in addition to other risk variables.

## Figures and Tables

**Figure 1 ijerph-17-00123-f001:**
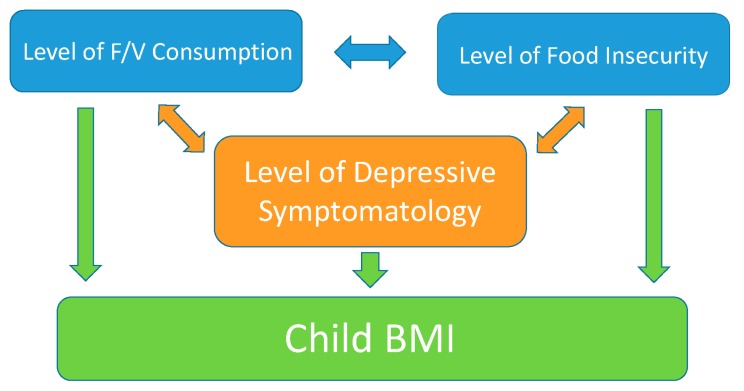
Conceptual Model Depicting the Relationship Between Maternal Depression and Obesity

**Table 1 ijerph-17-00123-t001:** Percent maternal depression screening items (*N* = 693) ^a,b^.

During the Past 2 Weeks, How Often:	Not at All (0)	Several Days (1)	More Than ½ the Days (2)	Nearly Every Day (3)
Have you been bothered by feeling down, depressed, or hopeless?	76.5%	17.0%	4.9%	1.6%
Have you been bothered by having little interest or pleasure in doing things?	82.1%	14.0%	2.9%	1.0%

^a^ Excludes cases with missing data on these items. ^b^ Items taken from the Patient Health Questionnaire—9 (Kroenke, Spitzer, & Williams, 2001).

**Table 2 ijerph-17-00123-t002:** Demographic characteristics by maternal depression level (*N* = 693).

Demographics	Depression Level			
	No Symptom Mothers ^a^ (*n* = 499) 72.0%	Low Symptom Mothers ^a^ (*n* = 154) 22.2%	High Symptom Mothers ^a^ (*n* = 40) 5.8%	Full Sample (*n* = 693)
Ethnicity				
White	22.2%	22.7%	20.0%	22.2%
Black	57.5%	50.0%	55.0%	55.7%
Hispanic	12.8%	18.2%	17.5%	14.3%
Other	7.4%	9.1%	7.5%	7.8%
Primary Caregiver Employed **	70.4%	65.1%	45.0%	67.7%
No other adult in the home	39.1%	37.3%	35.0%	38.5%
Has Partner/Married	56.3%	62.0%	50.0%	57.2%
Education (High School HS)				
No HS Diploma/No GED	11.6%	16.2%	15.0%	12.8%
HS Graduate/GED	37.6%	34.4%	42.5%	37.2%
Some Post Secondary Ed	50.8%	49.4%	42.5%	50.0%
Target Child is Male	49.2%	40.7%	38.5%	46.6%
Mean number of children living in the home (SD) ***	2.14 (1.07)	2.39 (1.23)	2.75 (1.37)	2.23 (1.14)

^a^ Summary scores on PHQ-2 of 0 = No Symptoms, 1 or 2 = Low-Level Symptoms, 3+ = High-Level Symptoms. Significant group differences, ** *p* < 0.01, *** *p* < 0.001.

**Table 3 ijerph-17-00123-t003:** Odds ratio for family food insecurity and inadequate child fruit/vegetable consumption in the home ^a^ by Maternal Depression.

Child Risk	Unadjusted Odds Ratio Low Depression Symptoms (95% CI)	Adjusted ^a^ Odds Ratio Low Depression Symptoms (95% CI)	Unadjusted Odds Ratio High Depression Symptoms (95% CI)	Adjusted ^a^ Odds Ratio High Depression Symptoms (95% CI)
Fruit/Vegetable Consumption	1.30 (0.87–1.92)	1.57 * (1.01–2.45)	2.53 * (1.97–5.84)	2.90 * (1.21–7.00)
Food Insecurity	2.07 ** (1.29–3.33)	2.14 ** (1.28–3.58)	6.36 *** (3.20–12.65)	7.81 *** (3.71–16.45)

^a^ Adjusted for maternal race/ethnicity, employment status, number of children in the home, level of education, presence of another adult in the home and child gender * *p* < 0.05; ** *p* < 0.01; *** *p* < 0.001.

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
