# Peer review of "Maternal Depression: Relationship to Food Insecurity and Preschooler Fruit/Vegetable Consumption"

_ijerph, 2019, doi:10.3390/ijerph17010123_

Round 1
Reviewer 1 Report
Clear presentation of your work. Important point that maternal depression is associated with a child's food intake. Some minor points I think would make for easier reading:
Abstract: Add some text to specific the 'implications' referred to, e.g. assessment of maternal depression in programs ...
Fig 1: perhaps refer to Child's BMI rather than pediatric patient as this study is not in a clinical setting where the children are 'patients'.
Line 82-86: Were there 2 interviews of each mother - one by educator and one by home visit Head Start staff?
If the one caregiver had 2 children in the program where they interviewed twice - once for each child - double counting of some mothers? Please clarify in text.
Line 172: "As seen in Table 1" - I can't see this in Table 1.
Table 3 & 4: label risk as inadequate F/V consumption
Table 4 title as F/V consumption - diet quality (overall) is not discussed here
Line 194: low level were twice as likely as what group?
Line 193-204: match text order to table - F/V first then FI
Discussion: first paragraph is a repeat of the Results not discussion of the results. I suggest deleting all except last sentence. Also no mechanism is suggested, it appears to be only descriptive results.
Author Response
We appreciate the reviewer’s thoughtful comments and offer a response for each requested edit below.
Reviewer 1
Clear presentation of your work. Important point that maternal depression is associated with a child's food intake.
Thank you for your kind comments.
Some minor points I think would make for easier reading:
Abstract: Add some text to specific the 'implications' referred to, e.g. assessment of maternal depression in programs.
Thank you for this feedback, we added a phrase to the abstract to expand on this idea.
Fig 1: perhaps refer to Child's BMI rather than pediatric patient as this study is not in a clinical setting where the children are 'patients'.
We made this edit, thank you for your insight.
Line 82-86: Were there 2 interviews of each mother - one by educator and one by home visit Head Start staff? If the one caregiver had 2 children in the program where they interviewed twice - once for each child - double counting of some mothers? Please clarify in text.
Each mother was interviewed once for each of her children in Head Start. However, anyone with multiple children, only one child’s data was used (the oldest) thus no duplications in the data set (n = 67 siblings removed). All interviews were conducted by a Head Start educator. We clarified this in the text.
Line 172: "As seen in Table 1" - I can't see this in Table 1.
Thank you for pointing this out. We have removed this language.
Table 3 & 4: label risk as inadequate F/V consumption
We have made the requested changes to Tables 3 and 4.
Table 4 title as F/V consumption - diet quality (overall) is not discussed here.
We have made this change here and throughout the paper.
Line 194: low level were twice as likely as what group?
We have clarified this comparison.
Line 193-204: match text order to table - F/V first then FI
We have reordered the text to match the table.
Discussion: first paragraph is a repeat of the Results not discussion of the results. I suggest deleting all except last sentence. Also no mechanism is suggested, it appears to be only descriptive results.
We have reduced the redundancy between the results and the first pgh of the discussion, only keeping enough overlap so that the reader can get a digestible summary of our results at the beginning of the discussion.
Reviewer 2 Report
This cross-sectional study examined the relationship of maternal depression with inadequate fruit/vegetable consumption and food insecurity of children in data collected from US Head Start centers. The authors found strong positive associations with both outcomes. The rationale for the study is explained, the analysis is adequate and the manuscript is well written. Nevertheless, I have a few important comments which needs to be considered before publication:
The abstract in current form is very brief and not informative at all. Further details need to be added, particularly regarding the methods. For example, please indicate what methods were used to collect data on maternal depression and the dietary outcomes, and what statistical method was used in the analysis. The fact that the study has a cross-sectional design also needs to be mentioned, and the geographical location (at least the country) should be also given. In the results, please provide the exact ORs and 95% CIs for the two examined associations. The conclusion of the abstract is very vague. Could the authors indicate in one sentence what implications they offer? It is not clear why the authors put so much emphasis on obesity in the Introduction. While it is true that the examined dietary factors may be related to this outcome, fruit/veg intake is much stronger connected with diet quality (as opposed to quantity), and food insecurity with undernutrition. I would suggest focusing more on these consequences rather than obesity. In the Introduction, or potentially in the Discussion, more thoughts should be given to the possibility that low socio-economic position (e.g. poverty) is a very plausible common cause of both maternal depression and inadequate diet. Perhaps it would worth considering adding this component to the conceptual model displayed in figure 1. In the Methods section, please indicate when (which year) the data collection took place, and it would be also useful to specify the location (i.e.: which Southern state?). If available, please indicate the response rate. Also, as one mother can have more than one child, information on how many children was collected overall? Table 1 would fit much better in the Results section as opposed to the Methods. Or alternatively, it can be moved to supplementary material. It is good that several potential confounders were taken into account in the statistical analysis (i.e. employment, education, ethnicity, etc.). Could you please give some information on how this data were collected? Were these questions part of the Family Map Inventory, or was it a separate questionnaire. In table 2, it would be good to show the exact p-values for every category instead of the asterisk labelling. It is quite confusing that the unadjusted (table 3) and adjusted (table 4) analyses were conducted with different statistical methods. I strongly suggest applying logistic regression to the unadjusted analysis as well, and combining the two tables with each other. By showing the unadjusted and adjusted ORs, the readers can see the impact of the covariates on the strength of the association. Please see the conventional way of displaying such results in tables in other similar studies. The discussion is clear and it is good that several methodological limitations of the study are considered. Potentially, it would be useful to put some more emphasis on the issue of the residual confounding by socio-economic position. In fact, this characteristic is probably a common cause rather than confounder in these associations, as I mentioned above. Although maternal education and employment was adjusted for, and the sample was rather restricted as most of them were at the lower end of the SEP spectrum, I believe this issue still plays an important role in the observed strong associations.Author Response
We appreciate the reviewer’s thoughtful comments and offer a response for each requested edit below.
This cross-sectional study examined the relationship of maternal depression with inadequate fruit/vegetable consumption and food insecurity of children in data collected from US Head Start centers. The authors found strong positive associations with both outcomes. The rationale for the study is explained, the analysis is adequate and the manuscript is well written.
Thank you for your kind comments.
Nevertheless, I have a few important comments which needs to be considered before publication:
The abstract in current form is very brief and not informative at all. Further details need to be added, particularly regarding the methods. For example, please indicate what methods were used to collect data on maternal depression and the dietary outcomes, and what statistical method was used in the analysis. The fact that the study has a cross-sectional design also needs to be mentioned, and the geographical location (at least the country) should be also given.
We have made these additions.
In the results [of the abstract], please provide the exact ORs and 95% CIs for the two examined associations.
We made these additions.
The conclusion of the abstract is very vague. Could the authors indicate in one sentence what implications they offer?
We have made this addition.
It is not clear why the authors put so much emphasis on obesity in the Introduction. While it is true that the examined dietary factors may be related to this outcome, fruit/veg intake is much stronger connected with diet quality (as opposed to quantity), and food insecurity with undernutrition. I would suggest focusing more on these consequences rather than obesity. In the Introduction, or potentially in the Discussion, more thoughts should be given to the possibility that low socio-economic position (e.g. poverty) is a very plausible common cause of both maternal depression and inadequate diet.
This study was instigated by the emerging conversation in pediatric obesity circles about the importance of maternal depression. This was a novel consideration that received national attention, and this context served as the foundation upon which we engaged in this project. In addition, FI has been found to be significantly related to both undernutrition and obesity but the incidence of the latter is of greater magnitude in our state, which ranks as one of the top 3 in obesity rates with higher rates across all socioeconomic levels. In fact, obesity rates have been documented to be higher among individuals in our state with food insecurity (42%) than among food secure individuals (33%) (Stuff, Casey, et al, 2007). However, you make a good point here (and below) that poverty is an important factor and needs to be incorporated in the paper. We have added a paragraph in the discussion.
Perhaps it would worth considering adding this component to the conceptual model displayed in figure 1.
As noted above we decided to edit the discussion to include a discussion about poverty and the impact of socioeconomic status on the variables investigated in our project. Since we did not change our intro, we did not add this to figure 1.
In the Methods section, please indicate when (which year) the data collection took place, and it would be also useful to specify the location (i.e.: which Southern state?). If available, please indicate the response rate. Also, as one mother can have more than one child, information on how many children was collected overall?
We edited the methods to include the year of data collection, the state, and the following information regarding mothers: Each mother was interviewed once for each of her children in Head Start. Howeve,r anyone with multiple children, only one child’s data was used (the oldest) thus no duplications in the data set (n = 67 siblings removed).
Table 1 would fit much better in the Results section as opposed to the Methods. Or alternatively, it can be moved to supplementary material.
We have moved Table 1 to the results section where we now report on maternal depression rates first before moving on to provide demographics and demographics my level of maternal depression.
It is good that several potential confounders were taken into account in the statistical analysis (i.e. employment, education, ethnicity, etc.). Could you please give some information on how this data were collected? Were these questions part of the Family Map Inventory, or was it a separate questionnaire.
Thank you for this feedback. It is accurate that all the confounding variable information was collected through the Family Map Interviews. We have added this clarification to the analysis plan section.
In table 2, it would be good to show the exact p-values for every category instead of the asterisk labelling.
We think the reviewer is asking for a P value in each category (no symptoms, low symptoms high). However, we conducted a three-way contrast and there is just an overall p.
It is quite confusing that the unadjusted (table 3) and adjusted (table 4) analyses were conducted with different statistical methods. I strongly suggest applying logistic regression to the unadjusted analysis as well, and combining the two tables with each other. By showing the unadjusted and adjusted ORs, the readers can see the impact of the covariates on the strength of the association. Please see the conventional way of displaying such results in tables in other similar studies.
We have made these edits in the text and the tables (now combined into table 3).
The discussion is clear and it is good that several methodological limitations of the study are considered.
Thank you for your kind comments.
Potentially, it would be useful to put some more emphasis on the issue of the residual confounding by socio-economic position. In fact, this characteristic is probably a common cause rather than confounder in these associations, as I mentioned above. Although maternal education and employment was adjusted for, and the sample was rather restricted as most of them were at the lower end of the SEP spectrum, I believe this issue still plays an important role in the observed strong associations.
We appreciate your perspective and agree. We added information in the discussion (and 3 additional references) related to this issue.
Round 2
Reviewer 2 Report
The authors have answered my comments appropriately and the quality of the manuscript has increased considerably as the result of the revision. I have only a few minor comments:
In the abstract, the authors should say that the study has a cross-sectional design. This is an essential information which is still missing. Also, please say that the presented ORs refer to the multivariable adjusted models. Finally, “logistic regression” should be written instead of “logistical regression”. I accept the answer which is given in the rebuttal letter in relation to my comments for the Introduction (“This study was investigated by…”). In my opinion, the Introduction section of the manuscript would benefit if this answer were incorporated in its text. The year of data collection is still missing in the Methods section. Please add.Author Response
Thank you for your review of our manuscript. In the abstract, we have added the reference to cross-sectional design and edited logistical regressions to logistic regressions as well as adding that the ORs presented are from the multivariable adjusted models. In the methods section of the paper, we added the year data was collected. In the introduction, we incorporated our response to reviewers into the text.
Thank you for your continued consideration of our manuscript.
The Authors